# Multipotent Cholinesterase Inhibitors for the Treatment of Alzheimer’s Disease: Synthesis, Biological Analysis and Molecular Docking Study of Benzimidazole-Based Thiazole Derivatives

**DOI:** 10.3390/molecules27186087

**Published:** 2022-09-18

**Authors:** Rafaqat Hussain, Hayat Ullah, Fazal Rahim, Maliha Sarfraz, Muhammad Taha, Rashid Iqbal, Wajid Rehman, Shoaib Khan, Syed Adnan Ali Shah, Sajjad Hyder, Majid Alhomrani, Abdulhakeem S. Alamri, Osama Abdulaziz, Mahmoud A. Abdelaziz

**Affiliations:** 1Department of Chemistry, Hazara University, Mansehra 21120, Pakistan; 2Department of Chemistry, University of Okara, Okara 56300, Pakistan; 3Department of Zoology, Wildlife and Fisheries, University of Agriculture Faisalabad, Sub-Campus Toba Tek Singh, Punjab 36050, Pakistan; 4Department of Clinical Pharmacy, Institute for Research and Medical Consultations (IRMC), Imam Abdulrahman Bin Faisal University, P.O. Box 1982, Dammam 31441, Saudi Arabia; 5Department of Agronomy, Faculty of Agriculture and Environment, The Islamia University of Bahawalpur Pakistan, Bahawalpur 63100, Pakistan; 6Faculty of Pharmacy, Universiti Teknologi MARA Cawangan Selangor Kampus Puncak Alam, Bandar Puncak Alam 42300, Selangor, Malaysia; 7Atta-ur-Rahman Institute for Natural Product Discovery (AuRIns), Universiti Teknologi MARA Cawangan Selangor Kampus Puncak Alam, Bandar Puncak Alam 42300, Selangor, Malaysia; 8Department of Botany, Government College Women University, Sialkot 51310, Pakistan; 9Department of Clinical Laboratories Sciences, The Faculty of Applied Medical Sciences, Taif University, Taif 21944, Saudi Arabia; 10Centre of Biomedical Sciences Research (CBSR), Deanship of Scientific Research, Taif University, Taif 21944, Saudi Arabia; 11Department of Chemistry, Faculty of Science, University of Tabuk, P.O. Box 741, Tabuk 71491, Saudi Arabia

**Keywords:** Synthesis, acetylcholinesterase, butyrylcholinesterase, benzimidazole, thiazole, structure-activity relationship, molecular docking

## Abstract

Twenty-four analogues of benzimidazole-based thiazoles (**1**–**24**) were synthesized and assessed for their in vitro acetylcholinesterase (AChE) and butyrylcholinesterase (BuChE) inhibitory potential. All analogues were found to exhibit good inhibitory potential against cholinesterase enzymes, having IC_50_ values in the ranges of 0.10 ± 0.05 to 11.10 ± 0.30 µM (for AChE) and 0.20 ± 0.050 µM to 14.20 ± 0.10 µM (for BuChE) as compared to the standard drug Donepezil (IC_50_ = 2.16 ± 0.12 and 4.5 ± 0.11 µM, respectively). Among the series, analogues **16** and **21** were found to be the most potent inhibitors of AChE and BuChE enzymes. The number (s), types, electron-donating or -withdrawing effects and position of the substituent(s) on the both phenyl rings B & C were the primary determinants of the structure-activity relationship (SAR). In order to understand how the most active derivatives interact with the amino acids in the active site of the enzyme, molecular docking studies were conducted. The results obtained supported the experimental data. Additionally, the structures of all newly synthesized compounds were elucidated by using several spectroscopic methods like ^13^C-NMR, ^1^H-NMR and HR EIMS.

## 1. Introduction

Alzheimer’s disease (AD) is a complex neurodegenerative and irretrievable abnormality. Its pathology involves the interactions of genetic risk and environmental factors [1]. This disease is considered age-related because it is the main cause of dementia, and is frequently found in old people. AD was counted as the fourth major death-causing disease in developed counties, after cancer, cerebral disease and cardiovascular disease [2]. Thus, it affects cholinergic neurons as well as cholinergic transmission. Moreover, AD is a multi-pathogenic illness; therefore, the recent strategy in drug discovery is to synthesize novel and potent anti-Alzheimer agents with significant inhibition potential for acetylcholinesterase (AChE) and butyrylcholinesterase (BuChE) enzymes [3]. Acetylcholine is crucial for cognitive processes, including memory. It lessens cholinergic neurotransmission in the brain, a factor that is significant in the cognitive decline brought on by Alzheimer’s disease [4,5]. BuChE is an enzyme related to AChE, and BuChE has been attracting growing attention due to its positive role in AD [6,7]. Butyrylcholinesterase prevents an enzyme from interfering with neurotransmitter transmission, which results in side effects for patients include nausea, vomiting, fever, and even death [8,9]. Thus, it might be considered as a significant tool for novel drug synthesis and used as the best target to treat AD. Therefore, the future development of potent BuChE inhibitors, as well as the regular practice of cholinesterase inhibitors, may lead to facilitating clinical results [10].

Benzimidazole possesses a diverse range of biological activities, such as anti-histaminic, anti-convulsant, anti-analgesic, proton pump inhibitors, anti-hypertensive, anti-cancer, anti-viral, anti-fungal and anti-coagulant actions [11,12,13,14,15,16,17]. There are some drug molecules such as benoxaprofen, albendazole, enviradine, bendamastin, omeprazole and astemizole that contain a benzimidazole moiety in their skeleton (Figure 1) [18].

Thiazoles scaffolds were recognized to play an essential role in the field of medicines. Thiazoles and their analogues were reported to have anti-Alzheimer’s, anti-diuretic and anti-bacterial activities [19].Moreover, thiazole-based hybrids have promising medicinal applications, including analgesic, anti-inflammatory [20], anti-microbial [21], anti-cancer [22], anti-hypoxic [23], anti-asthmatic [24] and anti-hypertensive activities [25]. In addition, thiazole derivatives and their chromene-based hybrids have demonstrated potent AChE-inhibitory activity [26,27,28,29].Furthermore, it was reported that due to the broad range of pharmacological significance of thiazole, a large number of commercially available drugs contain a thiazole nucleus in their structure, including abafungin, niridazole, tiazofurin, tiabendazole, ruvaconazole, vorelaxin and azereonam (Figure 2) [30,31].

Our research group had identified different classes of heterocyclic compounds as potent inhibitors [32,33,34,35,36,37,38,39,40,41]. Moreover, we had already reported on the anti-AChE and anti-BuChE activities of benzimidazole [42] and thiazole [43] derivatives (Figure 3). Keeping in view the biological significance of thiazole and benzimidazole skeleton containing analogues [44,45], here in this study, hybrid analogues of benzimidazole-based thiazole were designed and synthesized (**1**–**24**) as potent inhibitors of acetylcholinesterase and butyrylcholinesterase.

## 2. Results and Discussion

### 2.1. Chemistry

In this study, well-known methodology was adopted for the synthesis of benzimidazole-based thiazole derivatives (**1**–**24**) from readily accessible precursors such thiosemicarbazide, various phenacyl bromides and 2-marcaptobenzimidazole.

Initially, 2-bromoacetophenone (II) was reacted with 2-marcapto benzimidazole (I) in EtOH and Et_3_N (catalyst), and the reaction mixture was refluxed for 2–3 h to yield the first intermediate (III) [44]. An equivalent amount of intermediate (III) and thiosemicarbazide were further treated in EtOH and CH_3_COOH (glacial) and the solution was refluxed for about 4 h to yield a second intermediate product (IV). The reaction mixture was cooled to room temperature when it was finished, and the precipitate solid that resulted was then filtered and washed with n-hexane. The reaction’s progress had been monitored by using TLC plate. Finally, intermediate (IV) underwent cyclization with stirring overnight with a different substituted 2-bromoacetophenone in EtOH and Et_3_N (catalyst) to yield benzimidazole-based thiazole analogues (**1**–**24**) in moderate to good yield (Figure 1). The solvent was removed after being cooled to 25 °C and the resultant solid residue was then cleaned by washing with n-hexane and re-crystallized from ethyl acetate. The precise structures of all newly synthesized analogues were determined using several spectroscopic methods, including ^13^C-NMR, ^1^H-NMR and HR EIMS.

The ^1^H NMR spectrum of analogue **1** was recorded in DMSO-*d*_6_ on a Bruker 600 MHz instrument. The peak for the hydroxyl proton (OH) was observed at δ_H_ 10.11 (s, 1H, -OH). The most downfield singlets of two -NH protons, one for benzimidazole (NH) and another amino (NH) proton between the thiazole and benzimidazole rings, were resonated at δ_H_ 13.25 and 11.81, respectively. The molecule comprises three aromatic rings labeled as A, B C. Among the ring B protons, doublets appeared at δ_H_ 7.79 for two protons H-2′and H-6′ (Ar-H), while another two protons of this ring H-3′ H-5′ (Ar-H) were resonated at δ_H_7.88 (d, *J* = 8.88 Hz, 2H, Ar-H) as doublets, respectively. On the other hand, two aromatic protons labeled as H-3” and H-6“(Ar-H) of ring C resonated at δ_H_ 8.28 (s, 1H, H-3”) and 8.10 (s, 1H, H-6”) as singlets. Besides this, a triplet was observed for two aromatic protons of benzimidazole at δ_H_ 7.43 (t, J = 7.92 Hz, 2H, H-5/H-6). However, thiazole-H was resonated at δ_H_ 7.33 as a singlet. Moreover, two more aromatic protons of benzimidazole, namely H-4 and H-7, were also resonated as doublets at δ_H_7.27 (d, *J* = 7.68Hz, 1H, H-4) and 7.04 (d, *J* = 7.56Hz, 1H, H-7), respectively. Furthermore, a singlet was also observed for two active methylene protons (–CH_2_-) attached between sulfur and aromatic ring B at δ_H_2.34 (s, 2H, -S-CH_2_).

As for compound **1**, the ^13^C NMR signal δ_C-13_at 175.81 was attributed to a thiazole carbon (C) present between sulfur and nitrogen atoms, while two peaks at δ_C-13_139.85 (C-thiazole) and 110.66 (C-thiazole) were observed for the remaining two carbons of the thiazole ring. The peak at δ_C-13_138.96 (C) was observed for the benzimidazole carbon-bearing substitution, while two bridged carbons of benzimidazole were recorded at δ_C-13_137.27 (C). The carbon involved in the doublet with nitrogen (C=N) was resonated at δ_C-13_143.52 (C).The carbons of the aromatic phenyl rings B and C involving substitutions were resonated at δ_C-13_133.67 (C-OH), 132.00 (C-NO_2_), 127.83 (C), 126.25 (C-Cl) and 125.92 (C-Br). The peak corresponding to carbons of aromatic rings B and C without substitutions were resonated at δ_C-13_122.93 (CH-3′and CH-5′), 123.69 (CH), 120.66 (CH-2′and CH-6′) and 119.01 (CH), respectively. The peaks corresponding to the remaining four carbons of benzimidazole were observed at δ_C-13_ 123.7 (CH), 123.5 (CH), 115.8 (CH) and 115.3 (CH), respectively. The peak at δ_C-13_ 38.5 (-CH_2_-) corresponded to active methylene groups attached to a sulfur atom.

### 2.2. In Vitro Evaluation of AChE and BChE Inhibition by Novel Benzimidazole-Based Thiazoles

All the newly synthesized derivatives of benzimidazole-based thiazole (**1**–**24**) were evaluated as potential inhibitors of AChE and BChE, and IC_50_ was determined. All analogues showed moderate to good inhibitory potential against both AChE and BChE. The IC_50_ values ranged from 0.10 ± 0.05–11.10 ± 0.30 µM (for AChE) and 0.20 ± 0.050 µM–14.20 ± 0.10 µM (for BChE). Some of these compounds were shown to be more potent inhibitors than the standard drug Donepezil (IC_50_ = 2.16 ± 0.12 µM and 4.5 ± 0.11 µM respectively) (Table 1). The synthesized analogue was divided into four parts: benzimidazole ring A, thiazole portion, phenyl rings B & C, in order to better understand the structure-activity relationship (SAR). The nature, electron-withdrawing or -donating effects, number(s) and position of the substituent(s) on both phenyl rings B and C respectively were measured to be the key determinants of the structure-activity relationship (Figure 4). 

#### Structure–Activity Relationship of acetylcholinesterase and butyrylcholinesterase activities

The derivative **21** (IC_50_ = 0.10 ± 0.05µM and 0.20 ± 0.05 µM), bearing di-Cl groups at *meta*- and *para*-positions of both phenyl rings **B** and **C** correspondingly, was emerged as most effective inhibitors of both targeted enzymes (AChE & BuChE). Additionally, the analogue **16** (IC_50_ = 0.20 ± 0.050 µM for AChE) and 0.50 ± 0.050 µM for BuChE), which had a hydroxy group at the *para*-position of phenyl ring C and a nitro group at the *ortho*-position of phenyl ring B was found to be the 2nd most potent inhibitor of both AChE & BuChE enzymes. The different types, number (s), electron-donating or -withdrawing effect, positions of substituent(s) on the both phenyl rings B and C may account for the effective inhibitory potentials of analogues **16** and **21** (Figure 5). 

By comparing compounds **3–6** (Figure 6), we noticed that there is a difference in the inhibitory potential of these compounds in regard to the position of the substituent(s). This may be caused by the type, position, and number(s) of the substituent(s) on ring C.

Analogue **11** (dichloro group at *meta*- and *para*-positions on ring **C**, IC_50_ = 0.40 ± 0.050 µM and 1.10 ± 0.10 µM) was found to be superior to analogues **7** (nitro group at *ortho* on ring **C**, IC_50_ = 2.90 ± 0.10µM and 3.50 ± 0.10 µM) and **8** (nitro group at *meta* position on ring **C**, IC_50_ = 6.30 ± 0.10 µM and 7.90 ± 0.10 µM) (Figure 7).

By comparing compounds **13–16** (Figure 8), we noticed that there is a difference in the inhibitory potentials of these compounds in regard to the position of the substituent(s). This could be due to the nature of the substituent(s) on ring C.

By comparing compounds **17**–**19** and **21**–**22** (Figure 9), we noticed that there is a difference in the inhibitory potentials of these compounds in regard to the position of the substituent(s). This may be caused by the type, position, and number(s) of the substituent(s) on ring C.

Overall, it was found that the nature, electron-donating or -withdrawing effect, number and position of the substituent(s) on both rings B and C may considerably affect the inhibitory potentials of the synthesized analogues.

Galantamine is a phenanthrene that induces reversible inhibition of AChE-BuChE; Donepezil is a piperidine that causes reversible AChE inhibition, highly specific; and huperzine A is a pyridine that causes reversible AChE and specific inhibition. Under ideal test circumstances, each AChEI is ranked in the following order by its inhibitory efficacy (IC_50_) against the AChE activity: physostigmine (0.67 nM) > rivastigmine (4.4 nM) > Donepezil (6.7 nM) > TAK-147 (12 nM) > tacrine (77 nM) > ipidacrine (270 nM). Based on study conducted by Eisai scientists, derivatives with 4-aminopyridine such as ipidacrine and tacrine did not exhibit any selectivity, benzylpiperidine derivatives including TAK-147 and Donepezil exhibited high selectivity for AChE over BuChE, while the carbamate derivatives exhibited moderate selectivity. AChE inhibition by Donepezil is 40–500 times more effective than by galantamine, according to more recent studies. Galantamine leaves the brain more quickly than Donepezil does. Galantamine and Donepezil both inhibit brain AChE to a similar degree, according to their respective Ki values, which are 7.1–19.1 and 0.65–2.3 g/g in different species, respectively, the doses of needed for galantamine is 3–15 times greater than those of Donepezil [46].

### 2.3. Docking Study

The main objective of molecular docking study was to learn more about how newly afforded compounds bind to enzymes (i.e., both AChE and BuChE). Based on the residing co-crystal of each crystallographic structure, all of the compounds were docked. Each compound received a total of thirty conformations before the docking process. For additional protein-ligand interaction (PLI) profiling, the top-ranked conformations were chosen.

The docking results showed that all of the compounds were located in the active sites of both enzymes in the proper orientation. In general, we found that all the compounds had different substitution groups at their two ends, while the third end remained the same for all compounds, i.e., the electron-donating groups (also known as activated groups) and electron-withdrawing (also known as deactivated groups) at different positions. Additionally, we noticed that most active compounds held both the activated and deactivated groups at their two ends, which had a strong magnitude of activation. Surprisingly, the PLI profiles along with the in-vitro data revealed that compounds **16** and **21**, which contain activated and deactivated substituting groups over the benzene ring, had the highest inhibitory potential levels of the entire series. For example, compounds **16** and **21** have nearly identical activity levels against AChE but are ranked 1st **21** and 2nd **16** against BuChE, respectively.

Numerous significant interactions with catalytic residues were found in the comprehensive PLI profiles of both compounds against both targets. These interactions may have a significant impact on the improvement of the enzymatic activity of both enzymes. It was shown by PLI profile that analogue **21** adopted numerous key interactions with catalytic residues of AChE including the residues Phe330, Phe331, Tyr334, Asp72, Trp84, Tyr121 and Trp279 (Figure 10A), while this analogue against BuChE exhibited several important interactions such as Tyr332, Phe329, Ala328, Trp82, Asp70, Gly116, Gly117, Trp231, Ser287 and Leu286 (Figure 10B). The attached di-Cl on both ends of the compound may be the cause of its high potential, where the Cl that withdraw most of the electronic density from benzene ring, resulting in a partial positive charge over the benzene ring, causing this benzene ring to try to re-gain the stability via adopting several key interactions with active side residues, thereby enhancing the enzymatic activity. Overall, we observed that the compounds holding the substituted group, which had a strong magnitude of either withdrawal or donation, showed best potential against both targeted enzymes.

Similarly, the PLI profile shown by analogue **16** against AChE revealed several key interactions with the active site residues, including the residues Trp84, Glu199, Tyr121, Trp279, Asp285, Tyr334 and Asp72 (Figure 11C), while against the BuChE, this compound adopted several key interactions with Trp231, Leu286, Pro285, Phe329, Asp70, Ile69, Ala328 and Trp82 (Figure 11D). The elevated potential of this compound may be a result of the strong electron-donating and electron-withdrawing groups that are connected to it, i.e., hydroxyl and nitro group, where the -OH-substituting group donates some its electronic density with high potential to the 6c-ring and then further this electronic density cascades to other important moieties of the compounds; hence, In this manner, the overall potency of compound as an inhibitor against both targeted enzymes is high. In addition, the PLI profile revealed several key interactions, particularly with the side of the attached substituted groups or nearby moieties.

Additionally, the PLI profile shown by analogue **15** against AChE adopted several key interactions with the active site residues, including the residues Trp70, Trp84, Glu199, Phe331, Tyr334, Trp279 and Tyr70 (Figure 12E), while against the BuChE, this compound adopted several key interactions with Thr120, Asp70, Trp82, His438, Phe329, Trp231, Leu286 and Gly117 (Figure 12F). The elevated potential of this analogue **17** might have been due to the attached electron-withdrawing groups, such as di-Cl and nitro groups on both ends of analogue, where both di-Cl and –NO_2_ groups withdraw most of the electronic density from the benzene rings, resulting in a partial positive charge over the benzene ring, and further this benzene ring tries to re-gain the stability via adopting several key interactions with active side residues, thereby enhancing the enzymatic activity.

The calculated binding energies, number of hydrogen bonds, and number of closest residues surrounding the selected docked analogues into the active sites of both AChE and BuChE enzymes are shown in Table 2.

## 3. Experimental

### 3.1. General Information

All of the solvents and chemicals, with a purity range of 97 to 99%, were acquired from Sigma Aldrich. Using DMSO as the solvent, the NMR spectra were recorded using a Bruker Ultra Shield Plus NMR spectrometer. By using TMS as reference standard, the chemical sifts values were measured. The high-resolution mass spectra (electron impact, 60 eV) were run on a MAT-311A instrument (Germany). For visualization of the chromatograms, a UV lamp (Schimazdu, Germany) with a wavelength of 254/365 nm was used.

### 3.2. General Procedure for the Synthesis of Benzimidazole-Bearing 1,3-Thiazole Scaffolds (1–24)

The intermediate **(III)** was obtained by reacting phenacyl bromide (**II**, 1 mmol) with 2-marcapto benzimidazole (**I**, 1 mmol) in EtOH (10 mL) and Et_3_N (few drops) and stirred for 2-3h under reflux condition [44]. An amount of equivalent intermediate **(III)** and thiosemicarbazide were further treated in ethanol (10mL) and CH_3_COOH (glacial) and the solution was refluxed for about 4 h to yield the second intermediate product (**IV)**. Finally, intermediate (**IV)** underwent cyclization with stirring overnight with different substituted 2-bromoacetophenone in ethanol (10mL) Et_3_N (few drops) to yield benzimidazole-based thiazole analogues (**1**–**24**) in moderate to good yield. The solvent was removed after being cooled to room temperature and the resultant solid residue was then cleaned by being washed with n-hexane and re-crystallized from ethyl acetate. 

### 3.3. Spectral Analysis

#### 3.3.1. (E)-4-(2-(2-(2-((1H-benzo[d]imidazol-2-yl)thio)-1-(2-bromophenyl)ethylidene)hydrazinyl)thiazol-4-yl)-2-chloro-5-nitrophenol (**1**)

White crystals. Yield: 68% (0.33 g); ^1^H NMR (500 MHz, DMSO-*d_6_*): δ13.25 (s, 1H, -NH), 11.81 (s, -NH, 1H,), 10.11 (s, 1H, -OH),8.28 (s, 1H, H-3”), 8.10 (s, 1H, H-6”), 7.88 (d, *J*_(3′,2′/5′,6′)_ = 8.88 Hz, 2H, H-3′/H-5′), 7.79 (d, *J*_(2′,3′/6′,5′)_ = 8.46 Hz, 2H, H-2′/H-6′), 7.43 (t, *J*_(5(4,6)/6(5,7)_ = 7.92 Hz, 2H, H-5/H-6), 7.27 (d, *J*_(4,5)_ = 7.68 Hz, 1H, H-4), 7.11 (s, 1H, thiazole-H), 7.04 (d, *J*_(7,6)_ = 7.56 Hz, 1H, H-7), 2.34 (s, 2H, S-CH_2_); ^13^C-NMR (150 MHz, DMSO-d6): *δ* 175.8, 143.5, 139.8, 138.9, 137.2, 136.8, 133.6, 132.0, 127.8, 126.2, 125.9, 123.7, 123.5, 123.2, 122.9, 122.9, 122.0, 120.6, 120.6, 119.0, 115.8, 115.3, 110.6, 38.5.;HREI-MS: *m/z* [M+H]^+^calcd for C_24_H_17_BrClN_6_O_3_S_2_614.9664,found 614.9659. More information, please refer to the Appendix A.

#### 3.3.2. (E)-4-(2-(2-(2-((1H-benzo[d]imidazol-2-yl)thio)-1-(2-hydroxyphenyl)ethylidene)hydrazinyl)thiazol-4-yl)-2-chloro-5-nitrophenol (**2**)

White crystals. Yield: 64% (0.31 g); ^1^H NMR (500 MHz, DMSO-*d_6_*): δ13.39 (s, 1H, -NH), 13.27 (s, 1H, -NH), 12.01 (s, 1H, -OH), 10.15 (s, 1H, -OH), 8.91 (s, 1H, H-3”), 8.30 (s, 1H, H-6”),8.16–7.87 (m, 4H, H-4/H-5/H-6/H-7), 7.83–7.72 (m, 2H, H-3′/H-6′), 7.45 (t, *J*_(4′/3′,5′)_ = 7.74 Hz, 1H, H-4′), 7.25 (t, *J*_(5′/4′,3′)_ = 7.74 Hz, 1H, H-5′), 7.08 (s, 1H, thiazole-H), 2.28 (s, 2H, S-CH_2_); ^13^C-NMR (150 MHz, DMSO-d6): *δ* 183.9, 162.5, 158.4, 157.2, 155.3, 147.4, 146.4, 138.3, 137.2, 135.3, 132.4, 132.1, 131.5, 129.2, 127.5, 127.1, 126.4, 125.5, 124.6, 124.1, 121.0, 118.3, 117.0, 32.0.; HREI-MS: *m/z* [M+H]^+^calcd for C_24_H_18_ClN_6_O_4_S_2_553.0508,found 553.0502.

#### 3.3.3. (E)-4-(2-(2-(2-((1H-benzo[d]imidazol-2-yl)thio)-1-phenylethylidene) hydrazinyl)thiazol-4-yl)phenol(**3**)

White crystals. Yield: 69% (0.35 g); ^1^H NMR (500 MHz, DMSO-*d_6_*): δ12.35 (s, 1H, -NH), 11.78 (s, 1H, -NH), 9.70 (s, 1H, -OH), 7.96 (dd, *J*_(2′,3′/6′,5′)_ =8.1 Hz, *J*_(2′,4′/6′,4′)_ =2.0 Hz, 2H, H-2′/H-6′), 7.57–7.49 (m, 3H, H-3′/H-4′/H-5′), 7.43 (d, *J*_(2′,3′′/6′′,5′′)_ = 7.6 Hz, 2H, H-2′′/H-6′′), 6.94 (s, 1H, thiazole-H), 6.91 (d, *J*_(3′′,2′′/5′′,6′′)_ = 7.8 Hz, 2H, H-3′′/H-5′′), 3.56 (s, 2H, -S-CH_2_); ^13^C-NMR (125 MHz, DMSO-d6): δ 170.8, 157.6, 154.7, 149.3, 146.2, 138.7, 138.0, 133.7, 130.1, 128.0, 128.0, 127.9, 127.9, 127.3, 127.3, 124.7, 122.5, 122.3, 115.5, 115.5, 114.5, 114.3, 104.1, 36.6.; HREI-MS: m/z [M+H]^+^calcd for C_24_H_20_N_5_O_1_S_2_ 458.1106, found 458.1114.

#### 3.3.4. (E)-2-(2-(2-((1H-benzo[d]imidazol-2-yl)thio)-1-phenylethylidene)hydrazinyl)-4-(3-methoxyphenyl)thiazole (**4**)

Reddish crystals. Yield: 63% (0.30 g); ^1^H NMR (500 MHz, DMSO-*d_6_*): δ13.25 (s, 1H, -NH), 11.49 (s, 1H, -NH), 7.98 (dd, *J*_(2′,3′/6′,5′)_ =8.4 Hz, *J*_(2′,4′/6′,4′)_ =2.6 Hz, 2H, H-2′/H-6′), 7.60–7.55 (m, 3H, H-3′/H-4′/H-5′), 7.53–7.48 (m, 1H, H-6′′), 7.41 (t, *J*_(5′′/4′′,6′′)_ = 9.6 Hz, 1H, H-5′′), 7.32 (dd, *J*
_(2′′,6′′)_ = 2.7 Hz, *J*
_(2′′,4′′)_ = 1.8 Hz, 1H, H-2′′), 7.09(d, *J*_(4′′,5′′)_ = 8.3 Hz, 1H, H-4′′), 6.95 (s, 1H, thiazole-H), 3.58 (s, 2H, -S-CH_2_), 3.84 (s, 3H, -OCH_3_); ^13^C-NMR (125 MHz, DMSO-d6): δ 170.9, 160.3, 154.8, 149.4, 146.3, 138.3, 138.1, 133.2, 133.0, 130.2, 129.4, 128.0, 128.0, 127.4, 127.4, 122.4, 122.2, 119.0, 114.6, 114.4, 113.6, 112.8, 104.2, 55.0, 36.7.; HREI-MS: m/z [M+H]^+^calcd for C_25_H_22_N_5_O_1_S_2_ 472.1263, found 472.1271.

#### 3.3.5. (E)-2-(2-(2-((1H-benzo[d]imidazol-2-yl)thio)-1-phenylethylidene)hydrazinyl)-4-(p-tolyl)thiazole (**5**)

White crystals. Yield: 60% (0.28 g); ^1^H NMR (500 MHz, DMSO-*d_6_*): δ13.40 (s, 1H, -NH), 12.87 (s, 1H, -NH), 7.95 (dd, *J*_(2′,3′/6′,5′)_ = 7.0 Hz, *J*_(2′,4′/6′,4′)_ =2.4 Hz, 2H, H-2′/H-6′), 7.87 (d, *J*_(2′′,3′′/6′′,5′′)_ = 7.0 Hz, 2H, H-2′′/H-6′′), 7.59–7.52 (m, 3H, H-3′/H-4′/H-5′), 7.39 (d, *J*_(3′′,2′′/5′′,6′′)_ = 6.6 Hz, 2H, H-3′′/H-5′′), 6.73 (s, 1H, thiazole-H), 3.49 (s, 2H, -S-CH_2_), 2.37 (s, 3H, -CH_3_); ^13^C-NMR (125 MHz, DMSO-d6): δ 171.0, 154.9, 149.5, 146.4, 138.4, 138.2, 133.3, 131.0, 130.3, 129.3, 128.8, 128.8, 128.1, 128.1, 127.5, 127.5, 125.0, 125.0, 122.5, 122.3, 114.7, 114.5, 104.3, 36.8, 20.6.; HREI-MS: m/z [M+H]^+^calcd for C_25_H_22_N_5_S_2_ 456.1315, found 456.1322.

#### 3.3.6. (E)-2-(2-(2-((1H-benzo[d]imidazol-2-yl)thio)-1-phenylethylidene)hydrazinyl)-4-(3,4-dichlorophenyl)thiazole (**6**)

White crystals. Yield: 62% (0.28 g); ^1^H NMR (500 MHz, DMSO-*d_6_*): δ13.39 (s, 1H, -NH), 10.14 (s, 1H, -NH), 7.99 (dd, *J*_(2′,3′/6′,5′)_ = 8.7 Hz, *J*_(2′,4′/6′,4′)_ =2.5 Hz, 2H, H-2′/H-6′), 7.97 (d, *J*_(2′′,6′′)_ = 2.6 Hz, 1H, H-2′′), 7.82 (dd, *J*_(6′′,5′′)_ = 9.2 Hz, *J*_(6′′,2′′)_ = 1.9 Hz, 1H, H-6′′), 7.61 (d, *J*_(5′′,6′′)_ = 7.7 Hz, 1H, H-5′′), 7.58–7.53 (m, 3H, H-3′/H-4′/H-5′), 6.86 (s, 1H, thiazole-H), 3.69 (s, 2H, -S-CH_2_); ^13^C-NMR (125 MHz, DMSO-d6): δ 171.2, 155.1, 149.7, 146.6, 138.6, 138.4, 133.5, 132.9, 132.2, 132.0, 130.5, 130.2, 128.5, 128.3, 128.3, 127.7, 127.7, 126.5, 114.9, 122.7, 122.5, 114.7, 104.5, 37.0.; HREI-MS: m/z [M+H]^+^calcd for C_24_H_18_Cl_2_N_5_S_2_ 510.0376, found 510.0386.

#### 3.3.7. (E)-2-(2-(2-((1H-benzo[d]imidazol-2-yl)thio)-1-(p-tolyl)ethylidene)hydrazinyl)-4-(2-nitrophenyl)thiazole (**7**)

Yellow crystals. Yield: 71% (0.35 g); ^1^H NMR (500 MHz, DMSO-*d_6_*): δ13.66 (s, 1H, -NH), 10.12 (s, 1H, -NH), 8.10 (dd, *J*_(3′′,4′′)_ = 8.0 Hz, *J*_(3′′,5′′)_ = 2.8 Hz, 1H, H-3′′),8.07 (dd, *J*_(6′′,5′′)_ = 7.8 Hz, *J*_(6′′,4′′)_ = 1.6 Hz, 1H, H-6′′), 7.98–7.93 (m, 1H, H-5′′), 7.79 (t, *J*_(4′′/5′′,3′′)_ = 9.0 Hz, 1H, H-4′′), 7.73 (d, *J*_(2′,3′/6′,5′)_ = 7.8 Hz, 2H, H-2′/H-6′), 7.30 (d, *J*_(3′,2′/5′,6′)_ = 8.1 Hz, 2H, H-3′/H-5′), 6.89 (s, 1H, thiazole-H), 3.70 (s, 2H, -S-CH_2_), 2.44 (s, 3H, -CH_3_); ^13^C-NMR (125 MHz, DMSO-d6): δ 171.3, 155.2, 149.8, 148.4, 146.7, 140.3, 138.7, 138.5, 134.9, 132.2, 130.6, 129.2, 128.7, 128.7, 126.6, 126.6, 124.8, 124.0, 122.8, 122.6, 115.0, 114.8, 104.6, 37.1, 20.9.; HREI-MS: m/z [M+H]^+^calcd for C_25_H_21_N_6_O_2_S_2_ 501.1163, found 501.1172.

#### 3.3.8. (E)-2-(2-(2-((1H-benzo[d]imidazol-2-yl)thio)-1-(p-tolyl)ethylidene)hydrazinyl)-4-(3-nitrophenyl)thiazole (**8**)

White crystals. Yield: 65% (0.32 g); ^1^H NMR (500 MHz, DMSO-*d_6_*): δ12.42 (s, 1H, -NH), 11.90 (s, 1H, -NH), 8.63 (dd, *J*
_(2′′,6′′)_ = 2.5 Hz, *J*
_(2′′,4′′)_ = 1.6 Hz, 1H, H-2′′), 8.35–8.29 (m, 1H, H-6′′), 8.26–8.21 (m, 1H, H-4′′), 7.87 (t, *J*_(5′′/4′′,6′′)_ = 8.8 Hz, 1H, H-5′′), 7.75 (d, *J*_(2′,3′/6′,5′)_ = 7.7 Hz, 2H, H-2′/H-6′), 7.31 (d, *J*_(3′,2′/5′,6′)_ = 8.3 Hz, 2H, H-3′/H-5′), 6.87 (s, 1H, thiazole-H), 3.69 (s, 2H, -S-CH_2_), 2.42 (s, 3H, -CH_3_); ^13^C-NMR (125 MHz, DMSO-d6): δ 171.4, 155.3, 149.9, 148.1, 146.8, 140.4, 138.8, 138.6, 133.6, 133.3, 130.7, 130.3, 128.8, 128.8, 126.7, 126.7, 123.6, 122.9, 122.7, 122.4, 115.1, 114.9, 104.7, 37.2, 21.0.; HREI-MS: m/z [M+H]^+^calcd for C_25_H_21_N_6_O_2_S_2_ 501.1163, found 501.1172.

#### 3.3.9. (E)-4-(2-(2-(2-((1H-benzo[d]imidazol-2-yl)thio)-1-(3-hydroxy-2-nitrophenyl)ethylidene)hydrazinyl)thiazol-4-yl)-2-chloro-5-nitrophenol (**9**)

Yellow crystals. Yield: 73% (0.37 g); ^1^H NMR (500 MHz, DMSO-*d_6_*): δ13.39 (s, 1H, -NH), 13.30 (s, 1H, -NH), 12.21 (s, 1H, -OH),10.51 (s, 1H, -OH), 8.91 (s, 1H, H-3”), 8.35 (s, 1H, H-6”), 8.30 (d, *J*_(6′/5′)_ = 9.62 Hz, 1H, H-6′), 8.27–8.03 (m, 1H, H-4′), 7.91–7.85 (m, 1H, H-5′), 7.83 (d, *J*_(4,5/7,6)_ = 9.0 Hz, 2H, H-4/H-7), 7.76 (t, *J*_(5(4,6)/6(5,7))_ = 8.52 Hz, 2H, H-5/H-6), 7.10 (s, 1H, thiazole-H), 2.34 (s, 2H, S-CH_2_); ^13^C-NMR (150 MHz, DMSO-d6): *δ* 175.2, 162.1, 155.6, 143.1, 135.5, 135.6, 133.7, 131.7, 127.6, 127.3, 126.4, 126.3, 124.4, 123.8, 123.7, 121.0, 120.9, 120.7, 119.0, 118.8, 117.9, 112.4, 112.3, 30.4.; HREI-MS: *m/z* [M+H]^+^calcd for C_24_H_17_ClN_7_O_6_S_2_598.0365, found 598.0358.

#### 3.3.10. (E)-5-(2-(2-(2-((1H-benzo[d]imidazol-2-yl)thio)-1-([1,1’-biphenyl]-4-yl)ethylidene)hydrazinyl)thiazol-4-yl)-2-(dimethylamino)-4-nitrophenol (**10**)

White crystals. Yield: 60% (0.29 g);^1^H NMR (500 MHz, DMSO-*d_6_*): *δ* 13.64 (s, 1H, -NH), 11.54 (s, 1H, -NH), 10.12 (s, 1H, -OH),8.23 (s, 1H, H-3”), 8.18 (s, 1H, H-6”), 8.04 (d, *J*_(3,2/5,6)_ = 7.26 Hz, 4H, H-4/H-5/H-6/H-7), 7.99–7.92 (m, 4H, H-2′/H-3′/H-5′/H-6′), 7.64–7.59 (m, 5H, Ph-H), 7.03 (s, 1H, thiazole-H), 4.03 (s, 6H, -CH_3_), 2.51 (s, 2H, S-CH_2_); ^13^C-NMR (150 MHz, DMSO-d6): *δ* 188.5, 158.5, 157.3, 155.2, 148.3, 147.7, 143.0, 140.6, 139.4, 134.1, 133.8, 133.8, 130.0, 129.0, 128.9, 128.9, 127.9, 125.9, 125.9, 123.4, 121.3, 121.3, 120.6, 119.1, 119.1, 118.4, 118.4, 115.9, 115.2, 40.5, 40.5, 31.4.; HREI-MS: *m/z* [M+H]^+^calcd for C_32_H_28_N_7_O_3_S_2_622.1683, found 622.1678.

#### 3.3.11. (E)-2-(2-(2-((1H-benzo[d]imidazol-2-yl)thio)-1-(p-tolyl)ethylidene)hydrazinyl)-4-(3,4-dichlorophenyl)thiazole (**11**)

White crystals. Yield: 70% (0.34 g);^1^H NMR (500 MHz, DMSO-*d_6_*): δ12.53 (s, 1H, -NH), 11.69 (s, 1H, -NH), 7.97 (d, *J*_(2′′,6′′)_ = 2.5 Hz, 1H, H-2′′), 7.83 (dd, *J*_(6′′,5′′)_ = 9.5 Hz, *J*_(6′′,2′′)_ = 1.9 Hz, 1H, H-6′′), 7.79 (d, *J*_(2′,3′/6′,5′)_ = 8.5 Hz, 2H, H-2′/H-6′), 7.62 (d, *J*_(5′′,6′′)_ = 7.8 Hz, 1H, H-5′′), 7.38 (d, *J*_(3′,2′/5′,6′)_ = 8.6 Hz, 2H, H-3′/H-5′), 6.89 (s, 1H, thiazole-H), 3.58 (s, 2H, -S-CH_2_), 2.43 (s, 3H, -CH_3_),; ^13^C-NMR (125 MHz, DMSO-d6): δ 171.8, 155.7, 150.3, 147.2, 140.8, 139.0, 138.8, 133.5, 132.8, 132.6, 131.1, 130.8, 129.2, 129.2, 128.9, 127.3, 127.1, 127.1, 123.1, 122.9, 115.3, 115.1, 105.1, 37.6, 21.4.; HREI-MS: m/z [M+H]^+^calcd for C_25_H_20_Cl_2_N_5_S_2_ 524.0533, found 524.0542.

#### 3.3.12. (E)-4-(2-(2-(2-((1H-benzo[d]imidazol-2-yl)thio)-1-(4-methyl-2-nitrophenyl)ethylidene)hydrazinyl)thiazol-4-yl)-2-chloro-5-nitrophenol (**12**)

White crystals. Yield: 62% (0.30 g);^1^H NMR (500 MHz, DMSO-*d_6_*): *δ* 13.62 (s, 1H, NH), 13.38 (s, 1H, NH), 10.12 (s, 1H, -OH), 8.90 (s, 1H, H-3”), 8.23 (s, 1H, H-6”), 8.16 (s, 1H, H-3′), 8.02 (d, *J*_(5′,6′)_ = 5.58 Hz, 1H, H-5′), 7.87 (d, *J*_(6′,5′)_ = 8.46 Hz, 1H, H-6′), 7.75 (d, *J*_(3,2/5,6)_ = 8.4 Hz, 2H, H-4/H-7), 7.61 (t, *J*_(5(4,6)/6(5,7))_ = 8.34 Hz, 2H, H-5/H-6), 6.98 (s, 1H, thiazole-H), 2.50 (s, 2H, S-CH_2_), 1.91 (s, 3H, -CH_3_); ^13^C-NMR (150 MHz, DMSO-d6): *δ* 183.5, 162.1, 157.4, 155.3, 147.5, 147.2, 146.5, 141.4, 139.8, 137.0, 135.1, 133.8, 131.8, 129.9, 129.0, 126.8, 126.5, 125.4, 125.2, 124.4, 121.0, 118.8, 112.3, 30.5, 20.0.; HREI-MS: *m/z* [M+H]^+^ calcd for C_25_H_19_ClN_7_O_5_S_2_ 596.0567, found 596.0561.

#### 3.3.13. (E)-2-(2-(2-((1H-benzo[d]imidazol-2-yl)thio)-1-(2-nitrophenyl)ethylidene) hydrazinyl)-4-(p-tolyl)thiazole (**13**)

White microcrystals. Yield: 59% (0.27 g);^1^H NMR (500 MHz, DMSO-*d_6_*): δ13.40 (s, 1H, -NH), 10.95 (s, 1H, -NH), 8.15 (dd, *J*_(6′,5′)_ = 6.7 Hz, *J*_(6′,4′)_ = 2.5 Hz, 1H, H-6′), 8.04 (dd, *J*_(3′,4′)_ = 6.9 Hz, *J*_(3′,5′)_ =1.6 Hz, 1H, H-3′),7.93–7.89 (m, 1H, H-5′), 7.84 (d, *J*_(2′′,3′′/6′′,5′′)_ = 8.0 Hz, 2H, H-2′′/H-6′′), 7.66 (t, *J*_(4′/5′,3′)_ = 9.2 Hz, 1H, H-4′), 7.40 (d, *J*_(3′′,2′′/5′′,6′′)_ = 7.2 Hz, 2H, H-3′′/H-5′′), 6.97 (s, 1H, thiazole-H), 3.71 (s, 2H, -S-CH_2_), 2.34 (s, 3H, -CH_3_); ^13^C-NMR (125 MHz, DMSO-d6): δ 172.0, 155.9, 150.5, 147.4, 139.2, 139.0, 135.7, 134.6, 132.5, 132.2, 132.0, 130.3, 129.8, 129.8, 126.7, 126.0, 126.0, 125.7, 123.3, 123.1, 115.5, 115.3, 105.3, 37.8, 21.6.; HREI-MS: m/z [M+H]^+^calcd for C_25_H_21_N_6_O_2_S_2_ 501.1163, found 501.1172.

#### 3.3.14. (E)-2-(2-(2-((1H-benzo[d]imidazol-2-yl)thio)-1-(2-nitrophenyl)ethylidene) hydrazinyl)-4-(3-methoxyphenyl)thiazole (**14**)

White crystals. Yield: 58% (0.26 g);^1^H NMR (500 MHz, DMSO-*d_6_*): δ12.61 (s, 1H, -NH), 12.01 (s, 1H, -NH), 8.17 (dd, *J*_(6′,5′)_ = 6.5 Hz, *J*_(6′,4′)_ = 2.5 Hz, 1H, H-6′), 8.08 (dd, *J*_(3′,4′)_ = 6.8 Hz, *J*_(3′,5′)_ =1.8 Hz, 1H, H-3′),7.84–7.78 (m, 1H, H-5′), 7.68 (t, *J*_(4′/5′,3′)_ = 8.2 Hz, 1H, H-4′), 7.58–7.53 (m, 1H, H-6′′), 7.46 (t, *J*_(5′′/4′′,6′′)_ = 9.8 Hz, 1H, H-5′′), 7.35 (dd, *J*
_(2′′,6′′)_ = 2.0 Hz, *J*
_(2′′,4′′)_ = 2.4 Hz, 1H, H-2′′), 7.12(d, *J*_(4′′,5′′)_ = 8.3 Hz, 1H, H-4′′), 7.02 (s, 1H, thiazole-H), 3.86 (s, 3H, -OCH_3_), 3.80 (s, 2H, -S-CH_2_); ^13^C-NMR (125 MHz, DMSO-d6): δ 171.9, 161.3, 155.8, 150.4, 147.3, 139.1, 138.9, 135.6, 134.5, 134.2, 132.2, 132.0, 130.4, 126.6, 125.6, 123.2, 123.0, 120.0, 115.4, 115.2, 114.5, 113.8, 105.2, 56.0, 37.7.; HREI-MS: m/z [M+H]^+^calcd for C_25_H_21_N_6_O_3_S_2_ 517.1113, found 517.1123.

#### 3.3.15. (E)-2-(2-(2-((1H-benzo[d]imidazol-2-yl)thio)-1-(2-nitrophenyl)ethylidene) hydrazinyl)-4-(3,4-dichlorophenyl)thiazole (**15**)

White microcrystals. Yield: 63% (0.33 g);^1^H NMR (500 MHz, DMSO-*d_6_*): δ12.61 (s, 1H, -NH), 12.01 (s, 1H, -NH), 8.20 (dd, *J*_(6′,5′)_ = 7.0 Hz, *J*_(6′,4′)_ = 2.3 Hz, 1H, H-6′), 8.13 (dd, *J*_(3′,4′)_ = 7.4 Hz, *J*_(3′,5′)_ = 2.6 Hz, 1H, H-3′),8.01 (d, *J*_(2′′,6′′)_ = 1.6 Hz, 1H, H-2′′), 7.97–7.92 (m, 1H, H-5′), 7.85 (dd, *J*_(6′′,5′′)_ = 8.6 Hz, *J*_(6′′,2′′)_ = 1.7 Hz, 1H, H-6′′), 7.69 (t, *J*_(4′/5′,3′)_ = 9.5 Hz, 1H, H-4′), 7.64 (d, *J*_(5′′,6′′)_ = 7.7 Hz, 1H, H-5′′), 7.02 (s, 1H, thiazole-H), 3.80 (s, 2H, -S-CH_2_); ^13^C-NMR (125 MHz, DMSO-d6): δ 172.6, 156.5, 151.1, 148.0, 139.8, 139.6, 135.3, 135.1, 134.3, 133.6, 133.4, 132.7, 132.5, 131.6, 129.7, 127.9, 127.5, 126.3, 123.9, 123.7, 116.1, 115.9, 105.9, 38.4.; HREI-MS: m/z [M+H]^+^calcd for C_24_H_17_Cl_2_N_6_O_2_S_2_ 555.0227, found 555.0236.

#### 3.3.16. (E)-4-(2-(2-(2-((1H-benzo[d]imidazol-2-yl)thio)-1-(2-nitrophenyl)ethylidene) hydrazinyl) thiazol-4-yl)phenol (**16**)

White microcrystals. Yield: 62% (0.32 g);^1^H NMR (500 MHz, DMSO-*d_6_*): δ12.39 (s, 1H, -NH), 11.88 (s, 1H, -NH), 9.72 (s, 1H, -OH), 8.12 (dd, *J*_(6′,5′)_ = 7.6 Hz, *J*_(6′,4′)_ = 2.6 Hz, 1H, H-6′), 8.05 (dd, *J*_(3′,4′)_ = 7.9 Hz, *J*_(3′,5′)_ = 1.9 Hz, 1H, H-3′),7.79–7.74 (m, 1H, H-5′), 7.65 (t, *J*_(4′/5′,3′)_ = 9.2 Hz, 1H, H-4′), 7.42 (d, *J*_(2′′,3′′/6′′,5′′)_ = 7.2 Hz, 2H, H-2′′/H-6′′), 6.93 (d, *J*_(3′′,2′′/5′′,6′′)_ = 7.7 Hz, 2H, H-3′′/H-5′′), 6.75 (s, 1H, thiazole-H), 3.58 (s, 2H, -S-CH_2_); ^13^C-NMR (125 MHz, DMSO-d6): δ 172.1, 158.9, 156.0, 150.6, 147.5, 139.3, 139.1, 134.8, 134.6, 132.2, 132.0, 129.3, 129.3, 126.8, 126.0, 125.8, 123.4, 123.2, 116.8, 116.8, 115.6, 115.4, 105.4, 37.9.; HREI-MS: m/z [M+H]^+^calcd for C_24_H_19_N_6_O_3_S_2_ 503.0958, found 503.0965.

#### 3.3.17. (E)-2-(2-(2-((1H-benzo[d]imidazol-2-yl)thio)-1-(3,4-dichlorophenyl)ethylidene) hydrazinyl)-4-(2-nitrophenyl)thiazole (**17**)

White microcrystals. Yield: 64% (0.33 g);^1^H NMR (500 MHz, DMSO-*d_6_*): δ12.62 (s, 1H, -NH), 12.03 (s, 1H, -NH), 8.13 (dd, *J*_(6′′,5′′)_ = 8.2 Hz, *J*_(6′′,4′′)_ = 2.5 Hz, 1H, H-6′′), 8.06 (dd, *J*_(3′′,4′′)_ = 8.8 Hz, *J*_(3′′,5′′)_ = 2.2 Hz, 1H, H-3′′),7.97–7.93 (m, 1H, H-5′′), 7.89 (d, *J*_(2′,6′)_ = 2.3 Hz, 1H, H-2′), 7.85 (dd, *J*_(6′,5′)_ = 8.4 Hz, *J*_(6′,2′)_ =1.8 Hz, 1H, H-6′), 7.81 (t, *J*_(4′′/5′′,3′′)_ = 9.5 Hz, 1H, H-4′′), 7.73 (d, *J*_(5′,6′)_ =7.6 Hz, 1H, H-5′), 7.06 (s, 1H, thiazole-H), 3.81 (s, 2H, -S-CH_2_); ^13^C-NMR (125 MHz, DMSO-d6): δ 172.5, 156.4, 151.0, 149.2, 147.9, 139.7, 139.5, 136.5, 136.3, 136.1, 134.3, 133.4, 131.4, 131.1, 130.4, 127.1, 126.0, 125.2, 123.8, 123.6, 116.0, 115.8, 105.8, 38.3.; HREI-MS: m/z [M+H]^+^calcd for C_24_H_17_Cl_2_N_6_O_2_S_2_ 555.0227, found 555.0236.

#### 3.3.18. (E)-2-(2-(2-((1H-benzo[d]imidazol-2-yl)thio)-1-(3,4-dichlorophenyl)ethylidene) hydrazinyl)-4-(3-nitrophenyl)thiazole (**18**)

Reddish microcrystals. Yield: 61% (0.30 g);^1^H NMR (500 MHz, DMSO-*d_6_*): δ12.66 (s, 1H, -NH), 12.09 (s, 1H, -NH), 8.61 (dd, *J*
_(2′′,6′′)_ = 2.0 Hz, *J*
_(2′′,4′′)_ = 2.2 Hz, 1H, H-2′′), 8.32–8.27 (m, 1H, H-6′′), 8.21–8.14 (m, 1H, H-4′′), 7.95 (d, *J*_(2′,6′)_ = 2.5 Hz, 1H, H-2′), 7.89 (t, *J*_(5′′/4′′,6′′)_ = 9.0 Hz, 1H, H-5′′), 7.85 (dd, *J*_(6′,5′)_ = 8.7 Hz, *J*_(6′,2′)_ = 1.4 Hz, 1H, H-6′), 7.76 (d, *J*_(5′,6′)_ = 7.8 Hz, 1H, H-5′), 7.10 (s, 1H, thiazole-H), 3.83 (s, 2H, -S-CH_2_); ^13^C-NMR (125 MHz, DMSO-d6): δ 172.3, 156.2, 150.8, 149.0, 147.7, 139.5, 139.3, 136.3, 136.1, 134.5, 134.3, 134.1, 132.2, 131.2, 130.9, 126.9, 124.4, 123.6, 123.4, 123.2, 115.8, 115.6, 105.6, 38.1.; HREI-MS: m/z [M+H]^+^calcd for C_24_H_17_Cl_2_N_6_O_2_S_2_ 555.0227, found 555.0236.

#### 3.3.19. (E)-2-(2-(2-((1H-benzo[d]imidazol-2-yl)thio)-1-(3,4-dichlorophenyl)ethylidene) hydrazinyl)-4-(p-tolyl)thiazole (**19**)

Yellow crystals. Yield: 65% (0.36 g);^1^H NMR (500 MHz, DMSO-*d_6_*): δ12.59 (s, 1H, -NH), 11.97 (s, 1H, -NH), 7.97 (d, *J*_(2′,6′)_ = 2.0 Hz, 1H, H-2′), 7.87 (dd, *J*_(6′,5′)_ = 8.5 Hz, *J*_(6′,2′)_ = 1.5 Hz, 1H, H-6′), 7.83 (d, *J*_(2′′,3′′/6′′,5′′)_ = 7.8 Hz, 2H, H-2′′/H-6′′), 7.70 (d, *J*_(5′,6′)_ = 0.9 Hz, 1H, H-5′), 7.42 (d, *J*_(3′′,2′′/5′′,6′′)_ = 8.4 Hz, 2H, H-3′′/H-5′′), 6.99 (s, 1H, thiazole-H), 3.67 (s, 2H, -S-CH_2_), 2.41 (s, 3H, -CH_3_); ^13^C-NMR (125 MHz, DMSO-d6): δ 171.8, 155.7, 150.3, 147.2, 139.0, 138.8, 135.8, 135.6, 133.6, 131.9, 130.7, 130.4, 130.2, 129.7, 129.7, 126.4, 125.9, 125.9, 123.1, 122.9, 115.3, 115.1, 105.1, 37.6, 21.6.; HREI-MS: m/z [M+H]^+^calcd for C_25_H_20_Cl_2_N_5_S_2_ 524.0533, found 524.0542.

#### 3.3.20. (E)-2-(2-(2-((1H-benzo[d]imidazol-2-yl)thio)-1-(3,4-dichlorophenyl)ethylidene) hydrazinyl)-4-(3-methoxyphenyl)thiazole (**20**)

Reddish microcrystals. Yield: 61% (0.31 g);^1^H NMR (500 MHz, DMSO-*d_6_*): δ12.62 (s, 1H, -NH), 11.99 (s, 1H, -NH), 7.98 (d, *J*_(2′,6′)_ = 1.7 Hz, 1H, H-2′), 7.84 (dd, *J*_(6′,5′)_ = 8.6 Hz, *J*_(6′,2′)_ = 2.3 Hz, 1H, H-6′), 7.69 (d, *J*_(5′,6′)_ = 7.5 Hz, 1H, H-5′), 7.54–7.47 (m, 1H, H-6′′), 7.40 (t, *J*_(5′′/4′′,6′′)_ = 9.0 Hz, 1H, H-5′′), 7.29 (dd, *J*
_(2′′,6′′)_ = 2.9 Hz, *J*
_(2′′,4′′)_ = 1.4 Hz, 1H, H-2′′), 7.06(d, *J*_(4′′,5′′)_ = 8.5 Hz, 1H, H-4′′), 7.02 (s, 1H, thiazole-H), 3.83 (s, 3H, -OCH_3_), 3.78 (s, 2H, -S-CH_2_); ^13^C-NMR (125 MHz, DMSO-d6): δ 172.4, 161.8, 156.3, 150.9, 147.8, 139.6, 139.4, 136.4, 135.2, 134.7, 134.2, 131.3, 131.0, 130.9, 127.0, 123.7, 123.5, 120.5, 115.9, 115.7, 115.0, 114.3, 105.7, 56.5, 38.2.; HREI-MS: m/z [M+H]^+^calcd for C_25_H_20_Cl_2_N_5_OS_2_ 540.0483, found 540.0491.

#### 3.3.21. (E)-2-(2-(2-((1H-benzo[d]imidazol-2-yl)thio)-1-(3,4-dichlorophenyl)ethylidene) hydrazinyl)-4-(3,4-dichlorophenyl)thiazole (**21**)

Reddish microcrystals. Yield: 63% (0.34 g);^1^H NMR (500 MHz, DMSO-*d_6_*): δ12.65 (s, 1H, -NH), 12.11 (s, 1H, -NH), 8.03 (d, *J*_(2′′,6′′)_ = 2.8 Hz, 1H, H-2′′), 7.94 (d, *J*_(2′,6′)_ = 2.4 Hz, 1H, H-2′), 7.88 (dd, *J*_(6′′,5′′)_ = 9.0 Hz, *J*_(6′′,2′′)_ =1.5 Hz, 1H, H-6′′), 7.84 (dd, *J*_(6′,5′)_ = 8.8 Hz, *J*_(6′,2′)_ =2.6 Hz, 1H, H-6′), 7.74 (d, *J*_(5′,6′)_ =7.0 Hz, 1H, H-5′), 7.59 (d, *J*_(5′′,6′′)_ =7.3 Hz, 1H, H-5′′), 7.08 (s, 1H, thiazole-H), 3.86 (s, 2H, -S-CH_2_); ^13^C-NMR (125 MHz, DMSO-d6): δ 172.2, 156.1, 150.7, 147.6, 139.4, 139.2, 136.2, 136.0, 134.0, 133.9, 133.2, 133.0, 131.3, 131.1, 130.8, 129.3, 127.5, 126.8, 123.5, 123.3, 115.7, 115.5, 105.5, 38.0.; HREI-MS: m/z [M+H]^+^calcd for C_24_H_16_Cl_4_N_5_S_2_ 579.9568, found 579.9577.

#### 3.3.22. (E)-2-(2-(2-((1H-benzo[d]imidazol-2-yl)thio)-1-(3,4-dichlorophenyl)ethylidene) hydrazinyl)-4-(2-methoxyphenyl)thiazole (**22**)

White microcrystals. Yield: 68% (0.37 g);^1^H NMR (500 MHz, DMSO-*d_6_*): δ12.49 (s, 1H, -NH), 11.69 (s, 1H, -NH), 8.43 (dd, *J*_(6′′,5′′)_ = 7.6 Hz, *J*_(6′′,4′′)_ = 1.9 Hz, 1H, H-6′′), 7.91 (d, *J*_(2′,6′)_ = 2.9 Hz, 1H, H-2′), 7.82 (dd, *J*_(6′,5′)_ = 7.5 Hz, *J*_(6′,2′)_ =2.1 Hz, 1H, H-6′), 7.78 (d, *J*_(5′,6′)_ = 7.8 Hz, 1H, H-5′), 7.52 (t, *J*_(4′′/5′′,3′′)_ = 9.4 Hz, 1H, H-4′′), 7.43–7.37 (m, 1H, H-5′′), 7.18 (dd, *J*_(3′′,4′′)_ = 8.5 Hz, *J*_(3′′,5′′)_ = 1.5 Hz, 1H, H-3′′),7.03 (s, 1H, thiazole-H), 3.82 (s, 3H, -CH_3_), 3.59 (s, 2H, -S-CH_2_); ^13^C-NMR (125 MHz, DMSO-d6): δ 171.9, 157.5, 155.8, 150.4, 147.3, 139.1, 138.9, 135.9, 135.7, 133.7, 131.3, 130.8, 130.5, 129.9, 126.5, 123.2, 123.0, 121.7, 119.1, 115.4, 115.2, 111.3, 105.2, 56.3, 37.7.; HREI-MS: m/z [M+H]^+^calcd for C_25_H_20_Cl_2_N_5_OS_2_ 540.0483, found 540.0491.

#### 3.3.23. (E)-2-(2-(2-((1H-benzo[d]imidazol-2-yl)thio)-1-(3,4-dichlorophenyl)ethylidene) hydrazinyl)-4-(4-bromophenyl)thiazole (**23**)

Reddish microcrystals. Yield: 60% (0.29 g);^1^H NMR (500 MHz, DMSO-*d_6_*): δ12.44 (s, 1H, -NH), 11.73 (s, 1H, -NH), 7.93 (d, *J*_(2′,6′)_ = 2.8 Hz, 1H, H-2′), 7.87 (dd, *J*_(6′,5′)_ = 8.5 Hz, *J*_(6′,2′)_ = 1.7 Hz, 1H, H-6′), 7.79 (d, *J*_(2′′,3′′/6′′,5′′)_ = 7.4 Hz, 2H, H-2′′/H-6′′), 7.72 (d, *J*_(5′,6′)_ = 7.7 Hz, 1H, H-5′), 7.58 (d, *J*_(3′′,2′′/5′′,6′′)_ = 6.9 Hz, 2H, H-3′′/H-5′′), 6.88 (s, 1H, thiazole-H), 3.48 (s, 2H, -S-CH_2_); ^13^C-NMR (125 MHz, DMSO-d6): δ 171.5, 155.4, 150.0, 146.9, 138.7, 138.5, 135.5, 135.3, 133.3, 131.9, 131.9, 131.7, 130.4, 130.1, 128.1, 128.1, 126.1, 122.9, 122.7, 122.6, 115.0, 114.8, 104.2, 37.3.; HREI-MS: m/z [M+H]^+^calcd for C_24_H_17_BrCl_2_N_5_S_2_ 587.9484, found 587.9491.

#### 3.3.24. (E)-2-(2-(2-((1H-benzo[d]imidazol-2-yl)thio)-1-(p-tolyl)ethylidene)hydrazinyl)-4-(p-tolyl)thiazole (**24**)

Yellow crystals. Yield: 73% (0.37 g); ^1^H NMR (500 MHz, DMSO-*d_6_*): δ12.45 (s, 1H, -NH), 11.78 (s, 1H, -NH), 7.86 (d, *J*_(2′′,3′′/6′′,5′′)_ = 7.5 Hz, 2H, H-2′′/H-6′′), 7.72 (d, *J*_(2′,3′/6′,5′)_ = 6.8 Hz, 2H, H-2′/H-6′), 7.37 (d, *J*_(3′′,2′′/5′′,6′′)_ = 6.8 Hz, 2H, H-3′′/H-5′′), 7.34 (d, *J*_(3′,2′/5′,6′)_ = 6.9 Hz, 2H, H-3′/H-5′), 6.77 (s, 1H, thiazole-H), 3.56 (s, 2H, -S-CH_2_), 2.46 (s, 3H, -CH_3_), 2.38 (s, 3H, -CH_3_); ^13^C-NMR (125 MHz, DMSO-d6): δ 171.1, 155.9, 149.5, 146.5, 140.2, 138.5, 138.3, 131.1, 130.4, 129.3, 128.8, 128.8, 128.5, 128.5, 126.4, 126.4, 125.1, 125.1, 122.6, 122.4, 114.8, 114.6, 104.4, 36.9, 20.5, 20.6.; HREI-MS: *m/z* [M+H]^+^calcd for C_26_H_24_N_5_S_2_ 470.1469, found 470.1478.

### 3.4. Molecular Docking Protocol

The MOE software programme was used in molecular docking to determine how synthesized analogues interact with both targeted enzyme (AChE & BuChE) to triangulate the outcomes from in vitro and in silico analysis. Using the PDB codes 1ACL for AChE and 1P0P for BuChE, the RCSB protein databank’s crystal structures for both targets were retrieved. The crystallographic structures and all synthesized analogues were protonated using the default MOE-Dock module parameters, resulting in analogue structures and optimized enzyme. After this, a docking study was conducted using the optimized enzyme and analogues structures. Our earlier investigations contain all of the detailed information about the docking process [47,48,49].

### 3.5. Acetylcholinesterase Activity Assay Protocol

The inhibition of AChE and BChE was determined using a method described earlier [50,51]. Briefly, the stock solutions (1 mg/mL) of test analogues were prepared using DMSO. The working solutions (1–100 μg/mL) were prepared using serial dilutions (a serial dilution means a series of diluted solutions, e.g., 0.1 mg/mL, 0.2 mg/mL and so on; the solutions contained 5% DMSO and 95% water). The various concentrations of test compounds (10 μL) were pre-incubated with sodium phosphate buffer (0.1 M; pH 8.0; 150 μL) and AChE (0.1 U/mL; 20 μL) for 15 min at 25 °C. The reaction was initiated via the addition of DTNB (1 mM; 10 μL) and AChEI (1 mM; 10 μL). The mixture of reaction was mixed using a cyclomixer and incubated for 10 min at 25 °C. The absorbance was measured using a microplate reader at a 410 nm wavelength against the blank reading containing 10μL DMSO instead of the test compound (the solution contained 5% DMSO and 95% water). Utilizing the formula given in Equation (1), the percentage of inhibition was computed, and the IC_50_ was determined under the positive control of Donepezil (0.01–100 g/mL).

% Inhibition = (Absorbance of control—Absorbance of compound)/Absorbance of control × 100

IC_50_ is the concentration of a drug or inhibitor required to inhibit 50% of an enzyme’s activity which was calculated by constructing a non-linear regression graph between the percentages of inhibition vs. concentration, using Graph Pad prism software (version 5.3)(accessed on 5 August 2022). 

## 4. Conclusions

All of the newly synthesized derivatives of benzimidazole-based thiazole (**1**–**24**) were evaluated for inhibitory potentials (in vitro) against AChE and BuChE enzymes, respectively. Moreover, all derivatives (except derivative **10** due to its inactivity) displayed moderate to good inhibitory potentials in the range of 0.10 ± 0.05 to 11.10 ± 0.30 µM (for AChE) and 0.20 ± 0.050 µM to 14.20 ± 0.10 µM (for BuChE) under the positive control of Donepezil (IC_50_ = 2.16 ± 0.12 µM (for AChE) and 4.5 ± 0.11 µM (for BuChE)). Among the synthesized series, the analogue **21** (IC_50_ = 0.10 0.05 µM (for AChE) and 0.20 0.05 µM (for BuChE)) was identified as the most effective inhibitor of AChE and BuChE, even more potent than the standard, Donepezil.

## Data Availability

Not applicable.

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
