# Peer review of "Multipotent Cholinesterase Inhibitors for the Treatment of Alzheimer’s Disease: Synthesis, Biological Analysis and Molecular Docking Study of Benzimidazole-Based Thiazole Derivatives"

_molecules, 2022, doi:10.3390/molecules27186087_

Round 1
Reviewer 1 Report
The paper, entitled " Multipotent cholinesterase inhibitors for the treatment of Alzheimer’s disease: Synthesis, biological analysis and molecular docking study of benzimidazole based thiazole derivatives”, presents design and synthesis of benzimidazole based thiazole derivatives as cholinesterase inhibitors. The paper is well-organized and comprehensive. It will be useful for readers. The presented paper should be accepted after following minor revision:
1- The introduction, result and discussion parts and cited references should be improved adding information about various synthetic cholinesterase inhibitors, and also thiazole which is very popular linker in organic synthesis because of their wide-range activity and derivativable from different parts (by citing following articles:
Mekky, Ahmed EM et al, Synthetic Communications 51.21 (2021): 3332-3344. Demirayak, Åžeref, et al. Journal of Heterocyclic Chemistry 56.12 (2019): 3370-3386. Haroon, Muhammad, et al. Journal of Molecular Structure 1245 (2021): 131063. Hemaida, Aya Y., et al. ACS omega 6.29 (2021): 19202-19211.2- In Table 1, Ring A does not contain any substituent, “Ring A and Ring B” should be changed “Ring B and Ring C” as column header in correct form.
3- Selectivity index of synthesized compounds should be calculated and given in Table 1.
4- The spectrum of H, C NMR and MS should be provided in Supplementary Material file because the synthesized compounds are new.
Author Response
Reviewer-1:
Comments and Suggestions for Authors
The paper, entitled " Multipotent cholinesterase inhibitors for the treatment of Alzheimer’s disease: Synthesis, biological analysis and molecular docking study of benzimidazole based thiazole derivatives”, presents design and synthesis of benzimidazole based thiazole derivatives as cholinesterase inhibitors. The paper is well-organized and comprehensive. It will be useful for readers. The presented paper should be accepted after following minor revision:
1- The introduction, result and discussion parts and cited references should be improved adding information about various synthetic cholinesterase inhibitors, and also thiazole which is very popular linker in organic synthesis because of their wide-range activity and derivativable from different parts (by citing following articles:
Mekky, Ahmed EM et al, Synthetic Communications 51.21 (2021): 3332-3344. Demirayak, Åžeref, et al. Journal of Heterocyclic Chemistry 56.12 (2019): 3370-3386. Haroon, Muhammad, et al. Journal of Molecular Structure 1245 (2021): 131063. Hemaida, Aya Y., et al. ACS omega 6.29 (2021): 19202-19211.
Reply: The above mentioned articles were cited to improve the information about thiazole and various synthetics cholinesterase inhibitors as per kind suggestion.
2- In Table 1, Ring A does not contain any substituent, “Ring A and Ring B” should be changed “Ring B and Ring C” as column header in correct form.
Reply: In table 1 “Ring A & Ring B” was changed with “Ring B and Ring C” as per kind suggestion
3- Selectivity index of synthesized compounds should be calculated and given in Table 1.
Reply: The selectivity index of synthesized compounds were calculated and then incorporated in the table 1 as per kind suggestion
4- The spectrum of H, C NMR and MS should be provided in Supplementary Material file because the synthesized compounds are new.
Reply: Spectra’s of some representative compounds were provided as per kind suggestion
Reviewer 2 Report
Dear Authors,
The idea behind the paper is understandable, but English is very bad, the text is hard to understand, and it is unclear. IC50 is not an activity, the term is used incorrectly, IC50 represents the concentration of the tested compound which inhibits the 50% of enzyme activity. It should be written in the text and tables as IC50 not AChE inhibitory activity. You used DMSO, are you aware that it inhibits AChE and its content must be kept under 0.1% (DOI: 10.1021/acschemneuro.7b00344), also all controls should contain the corresponding % of DMSO as it is in the reaction mixture for IC50 determination. If the experiments were not done in such a way you should repeat them. The Methodology section on IC50 determination is unintelligibly written. Acetylcholinesterase should be written AChE, not AchE. Much has to be improved and additionally explained. You do not cite the literature on Donepezil IC50, is it in accordance with the IC50 you determined?
Author Response
Reviwer-2:
Comments and Suggestions for Authors
Dear Authors,
The idea behind the paper is understandable, but English is very bad, the text is hard to understand, and it is unclear.
Reply: Corrected according to the kind reviewer suggestion.
IC50 is not an activity, the term is used incorrectly, IC50 represents the concentration of the tested compound which inhibits the 50% of enzyme activity.
Reply: Corrected according to the kind reviewer suggestion.
It should be written in the text and tables as IC50 not AChE inhibitory activity.
Reply: Corrected according to the kind reviewer suggestion.
You used DMSO, are you aware that it inhibits AChE and its content must be kept under 0.1% (DOI: 10.1021/acschemneuro.7b00344), also all controls should contain the corresponding % of DMSO as it is in the reaction mixture for IC50 determination. If the experiments were not done in such a way you should repeat them.
Reply: Done the activity as per describe protocol with slight modification.
The Methodology section on IC50 determination is unintelligibly written.
Reply: Corrected according to the kind reviewer suggestion.
Acetylcholinesterase should be written AChE, not AchE.
Reply: The Acetylcholinesterase was written as (AChE) as per kind suggestion
Much has to be improved and additionally explained. You do not cite the literature on Donepezil IC50, is it in accordance with the IC50 you determined?
Reply: Literature about Donepezil was incorporated as per kind suggestion and the IC50 is in accordance that we have determined.
Reviewer 3 Report
1. Please provide copies of 1HNMR and 13CNMR of the products in supplementary material. I could not see the spectra. I could not give advices about the purity of the products. The authors must give all the spectra for review.
2. Please added some descriptive text about spectral data of the products to the manuscript.
3. For synthetic methods, the authors should add some related references.
4. The quality of the structures in the manuscript is not good. The all Schemes need to be homogenized throughout the manuscript.
5. According to the conclusions and aim of the article, binding energies for interaction of the synthesized compounds with both the targeted enzymes, Acetylcholinesterase (AChE) and Butyrylcholinesterase (BuChE) are necessary for this kind of study to support the conclusions.
6. Number H-bonds of the products with the related protein targets should collect in a Table.
7. The protein-ligand interaction (PLI) profiles for other compounds should provide in supplementary material.
8. The English of this manuscript is not good. The authors must check the English and rewrite all the manuscript.
Author Response
Reviwer-3:
Comments and Suggestions for Authors
- Please provide copies of 1HNMR and 13CNMR of the products in supplementary material. I could not see the spectra. I could not give advices about the purity of the products. The authors must give all the spectra for review.
Reply: The spectra’s of some representative compounds were provided as per kind suggestion
- Please add some descriptive text about spectral data of the products to the manuscript.
Reply: The descriptive text about spectral data of one of synthesized product was incorporated in the manuscript as per kind suggestion
- For synthetic methods, the authors should add some related references.
Reply: Incorporated according to the kind reviewer suggestion.
- The quality of the structures in the manuscript is not good. The all Schemes need to be homogenized throughout the manuscript.
Reply: The quality of the structures in the manuscript was improved and further all schemes were homogenized as per kind suggestion
- According to the conclusions and aim of the article, binding energies for interaction of the synthesized compounds with both the targeted enzymes, Acetylcholinesterase (AChE) and Butyrylcholinesterase (BuChE) are necessary for this kind of study to support the conclusions.
Reply: Binding energies for the interaction of the synthesized compounds with both targeted AChE & BuChE enzymes were incorporated in table-2 as per kind suggestion
- Number H-bonds of the products with the related protein targets should collect in a Table.
Reply: Number/s of hydrogen bonds formed by synthesized compounds with targeted protein were incorporated in the form of table-2 as per kind suggestion
- The protein-ligand interaction (PLI) profiles for other compounds should provide in supplementary material.
Reply: The protein ligand interaction (PLI) profile of moderate and least potent compounds were provided in supplementary material as per kind suggestion
- The English of this manuscript is not good. The authors must check the English and rewrite all the manuscript.
Reply: The mistakes related to English of manuscript was corrected as per kind suggestion
Round 2
Reviewer 2 Report
Dear Authors,
It is visible that you did some changes to the manuscript, but the English language still needs some editing. My comments are in the file attached.

Author Response
Reviewr-2:
Comments and Suggestions for Authors
Dear Authors,
It is visible that you did some changes to the manuscript, but the English language still needs some editing. My comments are in the file attached.
Reply: All the corrections are incorporated in the revised manuscript according the kind reviewer suggestion.
Reviewer 3 Report
Dear Editor
The paper was revised according to the reviewer’ comments.
In its current state it is ready for publication in your journal.
Best regards
Author Response
Reviewer-3:
Comments and Suggestions for Authors
Dear Editor
The paper was revised according to the reviewer’ comments.
In its current state it is ready for publication in your journal.
Reply: Thank you very much for such a nice wording.